# DeepACO: Neural-enhanced Ant Systems for Combinatorial Optimization

**Haoran Ye[1], Jiarui Wang[1], Zhiguang Cao[2,*], Helan Liang[1,*], Yong Li[3]**
[1] School of Computer Science & Technology, Soochow University
[2] School of Computing and Information Systems, Singapore Management University
[3] Department of Electronic Engineering, Tsinghua University
{hrye, jrwangfurffico}@stu.suda.edu.cn, zgcao@smu.edu.sg,
hlliang@suda.edu.cn, liyong07@tsinghua.edu.cn

## Abstract

Ant Colony Optimization (ACO) is a meta-heuristic algorithm that has been successfully applied to various Combinatorial Optimization Problems (COPs). Traditionally, customizing ACO for a specific problem requires the expert design of knowledge-driven heuristics. In this paper, we propose DeepACO, a generic framework that leverages deep reinforcement learning to automate heuristic designs. DeepACO serves to strengthen the heuristic measures of existing ACO algorithms and dispense with laborious manual design in future ACO applications. As a neural-enhanced meta-heuristic, DeepACO consistently outperforms its ACO counterparts on eight COPs using a single neural model and a single set of hyperparameters. As a Neural Combinatorial Optimization method, DeepACO performs better than or on par with problem-specific methods on canonical routing problems. Our code is publicly available at https://github.com/henry-yeh/DeepACO.

## 1 Introduction

Ant systems in nature are self-learners. They utilize chemical signals and environmental cues to locate and return food to the colony. The pheromone trails deposited by ants indicate the quality and distance of a food source. The intensity of pheromone trails increases as more ants visit and decreases due to evaporation, creating a self-learning foraging system.

Inspired by the ant systems in nature, researchers propose and develop Ant Colony Optimization (ACO) meta-heuristics for (but not limited to) Combinatorial Optimization Problems (COPs) [24, 26]. ACO deploys a population of artificial ants to explore the solution space through repeated solution constructions and pheromone updates. The exploration is biased toward more promising areas through instance-specific pheromone trails and problem-specific heuristic measures. Both the pheromone trail and the heuristic measure indicate how promising a solution component is. Typically, pheromone trails are initialized uniformly for all solution components and learned while solving an instance. On the contrary, heuristic measures are predefined based on prior knowledge of a problem, and devising proper heuristic measures for complicated COPs is quite challenging (an example is [49]).

Over the past decades, research and practice efforts have been dedicated to a careful design of heuristic measures in pursuit of knowledge-driven performance enhancement [57, 62, 33, 49, 26, 24]. However, this routine of algorithm customization exhibits certain deficiencies: 1) it requires extra effort and makes ACO less flexible; 2) the effectiveness of the heuristic measure heavily relies on expert knowledge and manual tuning; and 3) designing a heuristic measure for less-studied problems can be particularly challenging, given the paucity of available expert knowledge.

---

* Corresponding authors: Zhiguang Cao and Helan Liang.

37th Conference on Neural Information Processing Systems (NeurIPS 2023).

This paper proposes DeepACO, a generic neural-enhanced ACO meta-heuristic, and a solution to the above limitations. DeepACO serves to strengthen the heuristic measures of existing ACO algorithms and dispense with laborious manual design in future ACO applications. It mainly involves two learning stages. The first stage learns a problem-specific mapping from an instance to its heuristic measures by training neural models across COP instances. Guided by the learned measures, the second stage learns instance-specific pheromone trails while solving an instance with ACO. The heuristic measures learned in the first stage are incorporated into ACO (the second learning stage) by biasing the solution constructions and leading Local Search (LS) to escape local optima.

DeepACO is also along the line of recent progress in Neural Combinatorial Optimization (NCO) [12, 61, 7, 34]. Within the realm of NCO, DeepACO is more related to the methods that utilize heatmaps for algorithmic enhancement [32, 42, 90, 55, 42], but it is superior in its flexibility: DeepACO provides effective neural enhancement across eight COPs covering routing, assignment, scheduling, and subset problems, being the most broadly evaluated NCO technique to our knowledge. In addition, we propose three extended implementations for better balancing between exploitation and exploration: one featuring a multihead decoder, one trained with an additional top-$k$ entropy loss, and one trained with an additional imitation loss. They can be generally applied to heatmap-based NCO methods.

As a neural-enhanced version of ACO, DeepACO consistently outperforms its counterparts on eight COPs using a single neural model and a single set of hyperparameters after only minutes of training. As an NCO method, DeepACO performs better than or competitively against the state-of-the-art (SOTA) and problem-specific NCO methods on canonical routing problems while being more generalizable to other COPs. To the best of our knowledge, we are the first to exploit deep reinforcement learning (DRL) to guide the evolution of ACO meta-heuristics. Such a coupling allows NCO techniques to benefit from decades of ACO research (notably regarding theoretical guarantees [7, 26]), while simultaneously offering ACO researchers and practitioners a promising avenue for algorithmic enhancement and design automation.

In summary, we outline our **contributions** as follows:

- We propose DeepACO, a neural-enhanced ACO meta-heuristic. It strengthens existing ACO algorithms and dispenses with laborious manual design in future ACO applications.

- We propose three extended implementations of DeepACO to balance exploration and exploitation, which can generally be applied to heatmap-based NCO methods.

- We verify that DeepACO consistently outperforms its ACO counterparts across eight COPs while performing better than or on par with problem-specific NCO methods.

## 2 Related work

### 2.1 Neural Combinatorial Optimization

Neural Combinatorial Optimization (NCO) is an interdisciplinary field that tackles COPs with deep learning techniques. In general, existing NCO methods can be categorized into end-to-end and hybrid methods, and DeepACO belongs to the latter methodological category.

End-to-end methods in the former category learn autoregressive solution constructions or heatmap generation for subsequent sampling-based decoding. Within this realm, recent developments include better-aligned neural architectures [82, 64, 54, 50, 89, 13, 46], improved training paradigms [6, 56, 52, 83, 10, 66, 40, 45], advanced solution pipelines [47, 51, 48, 20, 78, 18, 65, 19], and broader applications [88, 95, 16, 29, 75, 9]. End-to-end methods are admirably efficient, but their constructed solutions can be further improved with iterative refinement and algorithmic hybridization.

Therefore, the hybrid methods in the latter category incorporate neural learners to make decisions within/for heuristics or generate heatmaps to assist heuristics. These methods either let neural learners make decisions within algorithmic loops [60, 87, 21, 17, 94, 86, 59], or generate heatmaps in one shot to assist subsequent algorithms. In the latter case, Xin et al. [90] propose to learn edge scores and node penalties to guide the searching process of LKH-3 [39], a highly optimized solver for routing problems. Kool et al. [55] train neural models to predict promising edges in routing problems, providing a neural boost for Dynamic Programming. Fu et al. [32] train small-scale GNN to build heatmaps for large TSP instances and feed the heatmaps into Monte Carlo Tree Search for solution

improvements. Hudson et al. [42] utilize neural models to generate regret in Guided Local Search for TSP. DeepACO is along the line of these works, but superior in terms of its flexibility, i.e., it provides effective neural enhancement across eight COPs covering routing, assignment, scheduling, and subset problems while the existing hybridized methods mostly focus on a limited set of routing problems.

## 2.2 Ant Colony Optimization

Ant Colony Optimization (ACO) is a meta-heuristic and evolutionary algorithm (EA) [4, 73] inspired by the behavior of ants in finding the shortest path between their colony and food sources [26].

The representative ACO meta-heuristics such as Ant System (AS) [25], Elitist AS [25], and MAX-MIN AS [76] provide general-purpose frameworks allowing for problem-specific customization. Such customization may involve designs of heuristic measures [49, 33], the incorporation of local search operators [3, 93], and the hybridization of different algorithms [44, 5]. DeepACO is not in competition with the most SOTA ACO algorithms for a specific COP. Instead, DeepACO can strengthen them with stronger heuristic measures and can in turn benefit from their designs.

ACO can utilize a group of techniques named hyper-heuristics [69] that are conceptually related to DeepACO. But hyper-heuristics mostly involve expert-designed heuristics to select from (heuristic selection hyper-heuristics) [27] or problem-specific and manually-defined components to evolve heuristics (heuristic generation hyper-heuristics) [14]. By comparison, DeepACO is more generic, requiring little prior knowledge of a COP. Its aim is not to compete with any hyper-heuristics; instead, for example, we can utilize hyper-heuristics to improve LS components in DeepACO.

Unlike the knowledge-driven ACO adaptations above, a recent method, namely ML-ACO [77], boosts the performance of ACO via Machine Learning (ML) for solution prediction. While ML-ACO provides an initial insight into coupling ML with ACO, it is preliminary and limited. ML-ACO tailors five features for Orienteering Problem and trains ML classifiers in a supervised manner. It entails high-demanding expert knowledge for feature designs and specialized solvers for optimal solutions, making itself inflexible and hard to scale. By comparison, DeepACO leverages DRL and demands little expert knowledge to apply across COPs.

## 3 Preliminary on Ant Colony Optimization

The overall ACO pipeline is depicted in Fig. 1. We begin with defining a COP model and a pheromone model. They are prerequisites for implementing ACO algorithms.

**COP model**   Generally, a COP model consists of, 1) a search space $\mathbf{S}$ defined over a set of discrete decision variables $X_i, i = 1, \ldots, n$, where each decision variable $X_i$ takes a value from a finite set $\boldsymbol{D}_i = \{v_i^1, \ldots, v_i^{|\boldsymbol{D}_i|}\}$; 2) a set of constraints $\Omega$ that the decision variables must satisfy; and 3) an objective function $f : \mathbf{S} \to \mathbb{R}_0^+$ to minimize. A feasible solution $\boldsymbol{s}$ to a COP is a complete assignment of all decision variables that satisfies all the constraints in $\Omega$.

**Pheromone model**   A COP model defines a pheromone model in the context of ACO. Without loss of generality, a pheromone model is a construction graph that includes decision variables as nodes and solution components as edges [26]. Each solution component $c_{ij}$, that represents the assignment of value $v_i^j$ to decision variable $X_i$, is associated with its pheromone trial $\tau_{ij}$ and heuristic measure $\eta_{ij}$. Both $\tau_{ij}$ and $\eta_{ij}$ indicate how promising it is to include $c_{ij}$ in a solution. Typically, ACO uniformly initializes and iteratively updates pheromone trails, but predefines and fixes heuristic measures.

As a motivating example, for TSP, a instance can be directly converted into a fully connected construction graph, where city $i$ becomes node $i$ and solution component $c_{ij}$ represents visiting city $j$ immediately after city $i$. Then, $\eta_{ij}$ is typically set to the inverse of the distance between city $i$ and $j$. After defining a COP model and a pheromone model, we introduce an ACO iteration, usually consisting of solution constructions, optional LS refinement, and pheromone updates.

**Solution construction and local search (optional)**   Biased by $\tau_{ij}$ and $\eta_{ij}$, an artificial ant constructs a solution $\boldsymbol{s} = \{s_t\}_{t=1}^n$ by traversing the construction graph. If an ant is located in node $i$ at the $t$-th construction step ($s_{t-1} = i$) and has constructed a partial solution $\boldsymbol{s}_{<t} = \{s_t\}_{t=1}^{t-1}$, the probability of

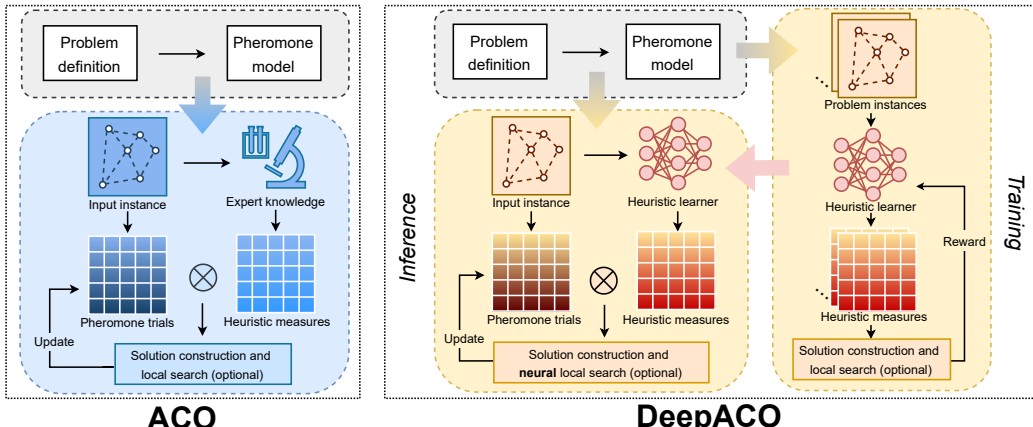

Figure 1: The schematic diagrams of ACO and DeepACO. DeepACO 1) additionally trains a heuristic learner across instances, 2) applies the well-trained heuristic learner to generate heuristic measures during inference, and 3) optionally leverages the learned heuristic measures to conduct local search interleaved with neural-guided perturbation.

selecting node $j$ as its next destination ($s_t = j$) is typically given by

$$P(s_t|\boldsymbol{s}_{<t}, \boldsymbol{\rho}) = \begin{cases} \dfrac{\tau_{ij}^{\alpha} \cdot \eta_{ij}^{\beta}}{\sum_{c_{il} \in \boldsymbol{N}(\boldsymbol{s}_{<t})} \tau_{il}^{\alpha} \cdot \eta_{il}^{\beta}} & \text{if } c_{ij} \in \boldsymbol{N}(\boldsymbol{s}_{<t}), \\ 0 & \text{otherwise.} \end{cases} \tag{1}$$

Here, $\boldsymbol{\rho}$ is a COP instance, $\boldsymbol{N}(\boldsymbol{s}_{<t})$ is the set of feasible solution components given the partial solution, and $\alpha$ and $\beta$ are the control parameters, which are consistently set to 1 in this work unless otherwise stated. To simplify the notations, we omit the dependence on $\boldsymbol{\rho}$ for $\tau$, $\eta$, $c$, and $\boldsymbol{N}$. Based on Eq. (1), constructing a complete solution requires an $n$-step graph traversal. The probability of generating $\boldsymbol{s}$ can be factorized as

$$P(\boldsymbol{s}|\boldsymbol{\rho}) = \prod_{t=1}^{n} P(s_t|\boldsymbol{s}_{<t}, \boldsymbol{\rho}). \tag{2}$$

After solution constructions, local search (LS) is optionally applied to refine the solutions.

**Pheromone update**   After solution constructions, the pheromone update process evaluates solutions and adjusts the pheromone trails accordingly, i.e., it increases the pheromone trails of components in the superior solutions while decreasing those in the inferior ones. The detailed update rules can differ depending on the ACO variation used.

ACO intelligently explores the solution space by iterating the above process, eventually converging on (sub)optimal solutions. We refer the readers to [26] for more details.

## 4   Methodology

DeepACO is schematically presented in Fig. 1 wherein a comparison is made with ACO. It dispenses with expert knowledge and learns a set of stronger heuristic measures to guide the ACO evolution. DeepACO involves parameterizing the heuristic space (Section 4.1), optionally interleaving LS with neural-guided perturbation (Section 4.2), and training a heuristic learner across instances (Section 4.3). Additionally, we introduce three extended designs (Section 4.4) to boost exploration.

### 4.1   Parameterizing heuristic space

We introduce a heuristic learner, defined by a graph neural network (GNN) with trainable parameters $\boldsymbol{\theta}$, to parameterize the heuristic space. The heuristic learner maps an input COP instance $\boldsymbol{\rho}$ to its

heuristic measures $\boldsymbol{\eta_\theta}(\boldsymbol{\rho})$, where we rewrite it as $\boldsymbol{\eta_\theta}$ for notation simplicity. It contains non-negative real values $\eta_{ij;\theta}$ associated with each solution component $c_{ij}, \forall i \in \{1, \ldots, n\}, \forall j \in \{1, \ldots, |\boldsymbol{D_i}|\}$. DeepACO constructs solutions following Eq. (2) but biased by $\boldsymbol{\eta_\theta}$:

$$P_{\boldsymbol{\eta_\theta}}(\boldsymbol{s}|\boldsymbol{\rho}) = \prod_{t=1}^{n} P_{\boldsymbol{\eta_\theta}}(s_t|\boldsymbol{s}_{<t}, \boldsymbol{\rho}). \tag{3}$$

In particular, we exploited the neural models recommended by Joshi et al. [48] and Qiu et al. [66]. It consists of a GNN backbone relying on anisotropic message passing and an edge gating mechanism, and a Multi-Layer Perceptron (MLP) decoder mapping the extracted edge features into heuristic measures. We defer the full details of this neural architecture to Appendix A.

## 4.2 Local search interleaved with neural-guided perturbation

In ACO, local search (LS) is optionally applied to refine the constructed solutions. However, LS is a myopic procedure in that it greedily accepts any altered solution with a lower objective value and easily gets trapped in local optima. In DeepACO, we intend the well-learned heuristic measures to indicate the global optimality of solution components. Leveraging such indicators and including solution components with greater global optimality can lead to better solutions eventually, if not immediately. Nevertheless, it is unrealistic to solely rely on the learned heuristic measures due to the inherent complexity of COPs.

Based on these considerations, we propose LS interleaved with neural-guided perturbation (NLS for short) in Algorithm 1. NLS interleaves LS aiming for a lower objective value and neural-guided perturbation biasing the learned optima. In each iteration, the first stage utilizes LS to repeatedly refine a solution until (potentially) reaching local optima. The second stage utilizes LS to slightly perturb the locally optimal solution toward gaining higher cumulative heuristic measures.

---

**Algorithm 1** NLS

1: **Input:** A solution $\boldsymbol{s}$; an objective function $f$; well-learned heuristic measures $\boldsymbol{\eta_\theta}$; a local search operator $LS$ that takes three inputs: a solution to refine, the targeted objective function, and the maximum iterations before encountering local optima; the number of perturbation moves $T_p$; the number of NLS iterations $T_{NLS}$
2: **Output:** The best improved solution $\boldsymbol{s}^*$
3: $\boldsymbol{s} = LS(\boldsymbol{s}, f, +\infty)$     // Improve $\boldsymbol{s}$ until reaching a local optimum
4: $\boldsymbol{s}^* = copy(\boldsymbol{s})$
5: **for** $iter = 1 \to T_{NLS}$ **do**
6:     $\boldsymbol{s} = LS(\boldsymbol{s}, \frac{1}{\boldsymbol{\eta_\theta}}, T_p)$ // Perturb $\boldsymbol{s}$ toward higher cumulative heuristic measures with $T_p$ moves
7:     $\boldsymbol{s} = LS(\boldsymbol{s}, f, +\infty)$
8:     $\boldsymbol{s}^* = \arg\min(f(\boldsymbol{s}^*), f(\boldsymbol{s}))$
9: **end for**

---

## 4.3 Training heuristic learner

We train the heuristic learner across COP instances. The heuristic learner $\boldsymbol{\theta}$ maps each instance $\boldsymbol{\rho}$ to its heuristic measures $\boldsymbol{\eta_\theta}$. Then, we minimize the expected objective value of both constructed solutions and NLS-refined constructed solutions:

$$\text{minimize} \quad \mathcal{L}(\boldsymbol{\theta}|\boldsymbol{\rho}) = \mathbb{E}_{\boldsymbol{s} \sim P_{\boldsymbol{\eta_\theta}}(\cdot|\boldsymbol{\rho})}[f(\boldsymbol{s}) + Wf(NLS(\boldsymbol{s}, f, +\infty))], \tag{4}$$

where $W$ is a coefficient for balancing two terms. Intuitively, the first loss term encourages directly constructing optimal solutions, which is, however, often held off by the complexity of COP. The second term encourages constructing solutions most fit for NLS, and it could be easier to learn high-quality solutions if coupling end-to-end construction with NLS. Even so, solely relying on the second term leads to inefficient training due to the small quality variance of the NLS-refined solutions. As a result, we find it useful to implement both terms for training. Note that the NLS process itself does not involve gradient flow.

In practice, we deploy Ant Systems to construct solutions stochastically following Eq. (3) for estimating Eq. (4). The pheromone trials are fixed to 1 to ensure an unbiased estimation. We apply a REINFORCE-based [85] gradient estimator:

$$\nabla\mathcal{L}(\boldsymbol{\theta}|\boldsymbol{\rho}) = \mathbb{E}_{\boldsymbol{s}\sim P_{\boldsymbol{\eta_\theta}}(\cdot|\boldsymbol{\rho})}[((f(\boldsymbol{s}) - b(\boldsymbol{\rho})) + W(f(NLS(\boldsymbol{s}, f, +\infty)) - b_{NLS}(\boldsymbol{\rho})))\nabla_{\boldsymbol{\theta}}\log P_{\boldsymbol{\eta_\theta}}(\boldsymbol{s}|\boldsymbol{\rho})],$$
(5)

where $b(\boldsymbol{\rho})$ and $b_{NLS}(\boldsymbol{\rho})$ are the average objective value of the constructed solutions and that of the NLS-refined constructed solutions, respectively.

## 4.4 Toward better exploration

The vanilla DeepACO fully exploits the underlying pattern of a problem and learns a set of aggressive heuristic measures (visualized in Appendix C.6) according to Eq. (4). Still, preserving exploration is of critical importance since COPs often feature many local optima [89]. To that end, we further present three extended designs to enable a better balance of exploration and exploitation. Note that they can generally be applied to heatmap-based NCO methods.

### 4.4.1 Multihead decoder

Multihead DeepACO implements $m$ MLP decoders on the top of the GNN backbone. It aims to generate diverse heuristic measures to encourage exploring different optima in the solution space. A similar multihead design has been utilized for autoregressive solution construction [89]. We extend this idea to the non-autoregressive heuristic generation here.

For training, Multihead DeepACO utilizes Eq. (4) calculated individually and independently for $m$ decoders and an extra Kullback-Leibler (KL) divergence loss expressed as

$$\mathcal{L}_{KL}(\boldsymbol{\theta}|\boldsymbol{\rho}) = -\frac{1}{m^2 n}\sum_{k=1}^{m}\sum_{l=1}^{m}\sum_{i=1}^{n}\sum_{j=1}^{|\boldsymbol{D}_i|}\tilde{\boldsymbol{\eta}}_{ij;\boldsymbol{\theta}}^{k}\log\frac{\tilde{\boldsymbol{\eta}}_{ij;\boldsymbol{\theta}}^{k}}{\tilde{\boldsymbol{\eta}}_{ij;\boldsymbol{\theta}}^{l}},$$
(6)

where $\tilde{\boldsymbol{\eta}}_{ij;\boldsymbol{\theta}}^{k}$ is the heuristic measure of the solution component $c_{ij}$ output by the $k$-th decoder after row-wise normalization.

In the inference phase, Multihead DeepACO deploys $m$ ant groups guided by their respective MLP decoders while the whole ant population shares the same set of pheromone trails.

### 4.4.2 Top-$k$ entropy loss

Entropy loss (reward) incentivizes agents to maintain the diversity of their actions. It is often used as a regularizer to encourage exploration and prevent premature convergence to suboptimal policies [68, 37, 51]. In the context of COP, however, most solution components are far from a reasonable choice for the next-step solution construction. Given this, we implement a top-$k$ entropy loss while optimizing Eq. (4), promoting greater uniformity only among the top $k$ largest heuristic measures:

$$\mathcal{L}_{H}(\boldsymbol{\theta}|\boldsymbol{\rho}) = \frac{1}{n}\sum_{i=1}^{n}\sum_{j\in\mathcal{K}_i}\bar{\boldsymbol{\eta}}_{ij;\boldsymbol{\theta}}\log(\bar{\boldsymbol{\eta}}_{ij;\boldsymbol{\theta}}).$$
(7)

Here, $\mathcal{K}_i$ is a set containing top $k$ solution components of decision variable $X_i$, and $\bar{\boldsymbol{\eta}}_{ij;\boldsymbol{\theta}}$ are the heuristic measures normalized within the set.

### 4.4.3 Imitation loss

If expert-designed heuristic measures are available, we can additionally incorporate an imitation loss, enabling the heuristic learner to mimic expert-designed heuristics while optimizing Eq. (4). It holds two primary merits. First, the expert-designed heuristic measures are less aggressive than the learned ones. So, it can function as a regularization to maintain exploration. Second, the heuristic learner can acquire expert knowledge. Perfecting the imitation first can guarantee that the learned heuristics are at least not inferior to the expert-designed counterparts.

Accordingly, the imitation loss is formulated as

$$\mathcal{L}_{I}(\boldsymbol{\theta}|\boldsymbol{\rho}) = \frac{1}{n}\sum_{i=1}^{n}\sum_{j=1}^{|\boldsymbol{D}_i|}\tilde{\boldsymbol{\eta}}_{ij}^{*}\log\frac{\tilde{\boldsymbol{\eta}}_{ij}^{*}}{\tilde{\boldsymbol{\eta}}_{ij;\boldsymbol{\theta}}},$$
(8)

where $\tilde{\eta}_{ij}^*$ and $\tilde{\eta}_{ij;\boldsymbol{\theta}}$ are expert-designed and learned heuristic measures, respectively, both after row-wise normalization.

## 5 Experimentation

### 5.1 Experimental setup

**Benchmarks** We evaluate DeepACO on eight representative COPs, including the Traveling Salesman Problem (TSP), Capacitated Vehicle Routing Problem (CVRP), Orienteering Problem (OP), Prize Collecting Traveling Salesman Problem (PCTSP), Sequential Ordering Problem (SOP), Single Machine Total Weighted Tardiness Problem (SMTWTP), Resource-Constrained Project Scheduling Problem (RCPSP), and Multiple Knapsack Problem (MKP). They cover routing, assignment, scheduling, and subset COP types. Their definitions and setups are given in Appendix D.

**Hardware** Unless otherwise stated, we conduct experiments on 48-core Intel(R) Xeon(R) Platinum 8350C CPU and an NVIDIA GeForce RTX 3090 Graphics Card.

### 5.2 DeepACO as an enhanced ACO algorithm

In this part, we do not apply NLS and set $W$ in Eq. (4) to 0 for training. It only entails minutes of training to provide substantial neural enhancement as shown in Fig. 2 (also see Appendix C.4). On the other hand, the extra inference time for DeepACO is negligible; for example, it takes less than 0.001 seconds for a TSP100.

**DeepACO for fundamental ACO algorithms** We implement DeepACO based on three fundamental ACO meta-heuristics: Ant System [25], Elitist Ant System [25], and MAX-MIN Ant System [76]. They are widely recognized as the basis for many SOTA ACO variations [44, 2, 5, 3, 93]. We compare the heuristic measures learned by DeepACO to the expert-designed ones (detailed separately for eight COPs in Appendix D) on 100 held-out test instances for each benchmark COP. The results in Fig. 2 show that DeepACO can consistently outperform its ACO counterparts, verifying the universal neural enhancement DeepACO can bring. We defer the results on more COP scales to Appendix C.1.

**DeepACO for advanced ACO algorithms** We also apply DeepACO to Adaptive Elitist Ant System (AEAS) [2], a recent ACO algorithm with problem-specific adaptations. In Fig. 3, we plot the evolution curves of AEAS based on the original and learned heuristic measures, respectively, and DeepACO shows clearly better performance. In light of this, we believe that it is promising to exploit DeepACO for designing new ACO SOTAs.

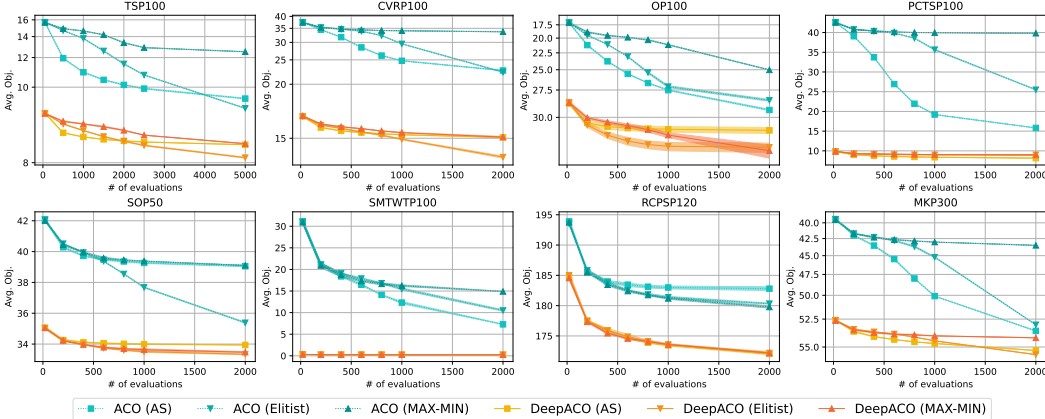

Figure 2: Evolution curves of fundamental DeepACO and ACO algorithms on eight different COPs. For each COP, we plot the best-so-far objective value (averaged over 100 held-out test instances and 3 runs) w.r.t. the number of used evaluations along the ACO iterations.

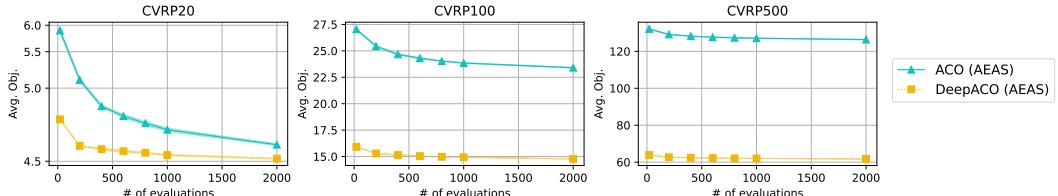

Figure 3: Evolution curves of DeepACO and ACO based on Adaptive elitist-ant system (AEAS) [2].

**DeepACO for different pheromone models**    The pheromone model used in ACO can vary, even for the same problem [63]. Following the naming convention in [63], we use $PH_{suc}$ to denote the pheromone model that captures the successively chosen items and $PH_{items}$ the one that directly captures how good each item is. We extend DeepACO from $PH_{suc}$ to $PH_{items}$ using a Transformer [79] encoder equipped with an MLP decoder. We showcase such flexibility of DeepACO on MKP in Fig. 7, where DeepACO using $PH_{items}$ can still outperform its ACO counterparts. It validates that DeepACO can be readily extended to variations of pheromone models using properly aligned neural models. We defer further discussions to Appendix A and C.5.

**DeepACO for better robustness to hyperparameter choice**    Increasing the robustness of ACO to hyperparameter choices has long been an important research topic [84, 92]. In Fig. 4, we adjust two important hyperparameters, i.e., *Alpha* that controls the transition probability of solution construction and *Decay* that controls the pheromone updates, evaluate DeepACO (AS) and ACO (AS) on the TSP100 test dataset, and plot their best objective values within the evolution of 4K evaluations. The results indicate that DeepACO is more robust to the hyperparameter choice given its much lower color variance. We thereby argue that data-driven training can also dispense with high-demanding expert knowledge for hyperparameter tuning.

**DeepACO for real-world instances**    We draw all 49 real-world symmetric TSP instances featuring *EUC_2D* and containing less than 1K nodes from TSPLIB. We infer instances with $n < 50$ nodes using the model trained on TSP20, those with $50 \leq n < 200$ nodes using the model trained on TSP100, and the rest using the model trained on TSP500. The evolution curves of DeepACO and ACO are shown in Fig. 5, suggesting that DeepACO can consistently outperform its ACO counterparts even when generalizing across scales and distributions.

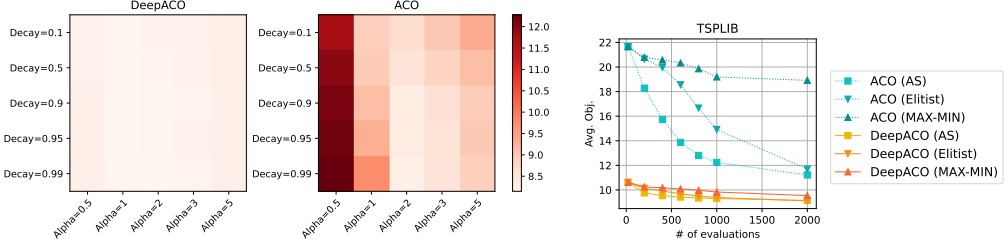

Figure 4: Sensitivity of DeepACO and ACO to the hyperparameter choice.

Figure 5: Evolution curves of DeepACO and ACO averaged over TSPLIB instances.

### 5.3   DeepACO as an NCO method

**Comparison on routing problems**    As shown in Table 1, DeepACO demonstrates competitive performance against the SOTA NCO methods for TSP. Here, we apply the generic 2-opt for NLS ($T_{NLS} = 10, T_p = 20$) and set $W$ in Eq. (4) to 9. Note that most NCO baselines are specialized for TSP or routing problems, while DeepACO is a general-purpose meta-heuristic and validated across eight different COPs. In addition, the results of ablation studies in Table 2 further validate our designs; that is, both neural-guided perturbation and training with LS effectively strengthen DeepACO. More results on TSP100 and CVRP are given in Appendix C.2.

Table 1: Comparison results of DeepACO and NCO baselines. We report per-instance execution time.

| Method | TSP500 | | | TSP1000 | | |
| --- | --- | --- | --- | --- | --- | --- |
| | Obj. | Gap(%) | Time(s) | Obj. | Gap(%) | Time(s) |
| LKH-3 [39] | 16.55 | 0.00 | 2.6 | 23.12 | 0.00 | 11 |
| ACO | 17.57 | 6.16 | 7.3 | 24.94 | 7.87 | 54 |
| AM* [54] | 21.46 | 29.7 | 0.6 | 33.55 | 45.1 | 1.5 |
| POMO* [56] | 20.57 | 24.4 | 0.6 | 32.90 | 42.3 | 4.1 |
| GCN+MCTS* [32] | 16.97 | 2.54 | 50 | 23.86 | 3.22 | 100 |
| DIMES+MCTS [66] | 17.01 | 2.78 | 11 | 24.45 | 5.75 | 32 |
| DIFUSCO+MCTS* [78] | 16.63 | 0.46 | 4.7 | 23.39 | 1.17 | 11 |
| SO* [18] | 16.94 | 2.40 | 15 | 23.77 | 2.80 | 26 |
| Pointerformer [46] | 17.14 | 3.56 | 1.4 | 24.80 | 7.30 | 4.0 |
| H-TSP* [65] | - | - | - | 24.65 | 6.62 | 0.3 |
| DeepACO $\{T = 2\}$ | 16.94 | 2.36 | 2.0 | 23.99 | 3.76 | 6.4 |
| DeepACO $\{T = 10\}$ | 16.86 | 1.84 | 10 | 23.85 | 3.16 | 32 |

Results of methods with * are drawn from [18, 65, 78].

Table 2: Ablation studies of DeepACO. We additionally evaluate DeepACO 1) with vanilla LS and more iterations, 2) implementing random perturbation for LS, and 3) trained with $W = 0$ for Eq. (4).

| DeepACO | TSP500 | | TSP1000 | |
| --- | --- | --- | --- | --- |
| | Obj. | Time(s) | Obj. | Time(s) |
| w/ vanilla LS | 16.90 | 17 | 23.89 | 65 |
| w/ random perturbation | 16.92 | 11 | 23.90 | 39 |
| w/ $\{W = 0\}$ for training | 17.05 | 10 | 24.23 | 32 |
| Original | 16.86 | 10 | 23.85 | 32 |

**For heatmaps with better exploration**   We implement the three extended designs introduced in Section 4.4 without LS and compare their evolution curves with vanilla DeepACO in Fig. 6. We apply 4 decoder heads for Multihead DeepACO and set $K$ to 5 for the top-K entropy loss. The coefficients for the additional loss terms (i.e., $\mathcal{L}_{KL}$, $\mathcal{L}_H$, and $\mathcal{L}_I$) are set to 0.05, 0.05, and 0.02, respectively. The results indicate that these designs lead to better performance on TSP20, 50, and 100 since small-scale COPs typically desire more exploration.

**For enhancing heatmap decoding with pheromone update**   Though we can conduct pure solution samplings similar to [47, 66, 78] given the learned heuristic measures, we demonstrate the superiority of ACO evolution (without NLS) with self-learning pheromone update in Fig. 8. Furthermore, coupling heatmaps with ACO can benefit from extensive ACO research, such as advanced ACO variations, algorithmic hybridizations, and seamless incorporation of local search operators.

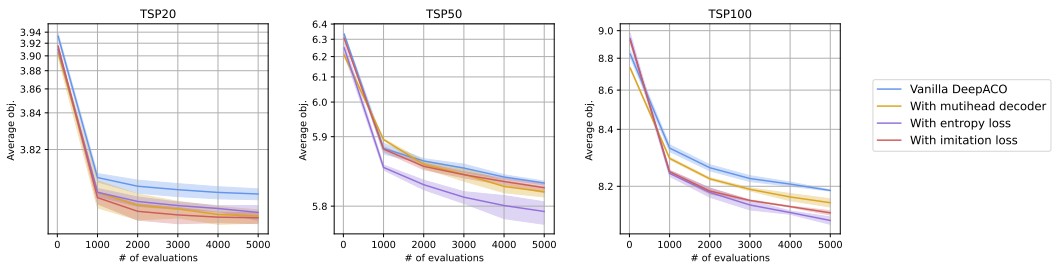

Figure 6: Evolution curves of vanilla DeepACO and its three extended implementations on TSP20, 50, and 100. The results are averaged over 3 runs on held-out test datasets each with 100 instances.

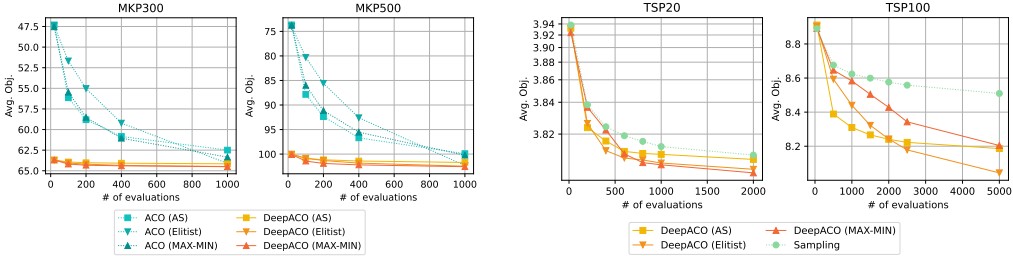

Figure 7: Evolution curves of DeepACO and ACO on MKP300 and MKP500 using the pheromone model $PH_{items}$.

Figure 8: Solution comparison between using Ant System evolution and pure sampling.

# 6   Conclusion and limitation

In this work, we propose DeepACO to provide universal neural enhancement for ACO algorithms and automate heuristic designs for future ACO applications. We show that DeepACO performs

consistently better than its ACO counterparts and is on par with the specialized NCO methods. For the Operations Research (OR) community, DeepACO demonstrates a promising avenue toward leveraging deep reinforcement learning for algorithmic enhancement and design automation. For the Machine Learning (ML) community, DeepACO presents a versatile and adaptable NCO framework that can seamlessly integrate with SOTA ACO and EA techniques, including improved rules for solution construction and pheromone update [76, 11], algorithmic hybridizations [67, 91], and incorporation of sophisticated local search operators [3, 93].

However, DeepACO is potentially limited by compressing all learned heuristic information into an $n \times n$ matrix of heuristic measures. Due to the complexity of COPs and this restricted way of expressing problem patterns, DeepACO may fail to produce close-to-optimal solutions when not incorporating LS components. To address such limitations, we plan to investigate 3D or dynamic heuristic measures in our future work.

## Acknowledgments and Disclosure of Funding

The authors appreciate the helpful discussions with Yu Hong, Juntao Li, and anonymous reviewers. Yu Hu, Jiusi Yin, and Tao Yu also contributed to this work. This work was supported by the National Natural Science Foundation of China (NSFC) [grant numbers 61902269, 52074185]; the National Key R&D Program of China [grant number 2018YFA0701700]; the Undergraduate Training Program for Innovation and Entrepreneurship, Soochow University [grant number 202210285001Z, 202310285041Z]; the Priority Academic Program Development of Jiangsu Higher Education Institutions, China; and Provincial Key Laboratory for Computer Information Processing Technology, Soochow University [grant number KJS1938].

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

# DeepACO: Neural-enhanced Ant Systems for Combinatorial Optimization (Appendix)

## A Neural architecture

### A.1 GNN

For pheromone model $PH_{suc}$, we employ the neural architecture recommended by Joshi et al. [48] and Qiu et al. [66] as the GNN backbone, which relies on an anisotropic message passing scheme and an edge gating mechanism. The GNN we use has 12 layers. For the $l$-th layer, we denote the node feature of $v_i$ as $\mathbf{h}_i^l$ and the edge feature of edge $< i, j >$ as $\mathbf{e}_{ij}^l$. The propagated features are defined as

$$\mathbf{h}_i^{l+1} = \mathbf{h}_i^l + \alpha(\mathrm{BN}(\mathbf{U}^l\mathbf{h}_i^l + \mathcal{A}_{j \in \mathcal{N}_i}(\sigma(\mathbf{e}_{ij}^l) \odot \mathbf{V}^l\mathbf{h}_j^l))), \tag{9}$$

$$\mathbf{e}_{ij}^{l+1} = \mathbf{e}_{ij}^l + \alpha(\mathrm{BN}(\mathbf{P}^l\mathbf{e}_{ij}^l + \mathbf{Q}^l\mathbf{h}_i^l + \mathbf{R}^l\mathbf{h}_j^l)), \tag{10}$$

where $\mathbf{U}^l, \mathbf{V}^l, \mathbf{P}^l, \mathbf{Q}^l, \mathbf{R}^l \in \mathbb{R}^{d \times d}$ are learnable parameters, $\alpha$ denotes the activation function, BN denotes batch normalization [43], $\mathcal{A}_{j \in \mathcal{N}_i}$ denotes the aggregation operation over the neighbourhood of $v_i$, $\sigma$ is the sigmoid function, and $\odot$ is the Hadamard product. In this paper, we define $\alpha$ as SiLU [28] and $\mathcal{A}$ as mean pooling.

We map the extracted edge features into real-valued heuristic measures using a 3-layer MLP with skip connections [38] to the embedded node feature $\mathbf{h}_i^{12}$. We apply SiLU [28] as the activation function except for the output layer, where the sigmoid function is utilized to obtain normalized outputs.

### A.2 Transformer

For pheromone model $PH_{items}$, the architecture mostly follows the transformer encoder ($num\_hidden\_layers = 3, hidden\_size = 32, num\_attention\_heads = 2$) but deprecates its positional encoding. On top of it, we add position-wise feedforward layers ($num\_hidden\_layers = 3, hidden\_size = 32$) that map the hidden representations of each solution component into its real-valued heuristic measure. Overall, it is similar to the neural networks used in [23, 35].

## B Details of NLS

In NLS, neural-guided perturbation does not involve a new neural network other than that introduced in Section 4.1. It directly leverages the learned heuristic measures to guide the perturbation process. Specifically, each time a solution reaches local optima, it gets perturbed with local search (LS) which iteratively maximizes the heuristic measures of its solution components. Note that NLS can be implemented using an arbitrary LS operator.

Take TSP as an example. With 2-opt as the LS operator, NLS alternates between (a) optimizing a solution to minimize its tour length and (b) perturbing it to maximize the total heuristic measures of its edges. The two processes are similar, except that we use the inverse of heuristic measures in (b) while the distance matrix in (a) as indicators of good edges (line 6 in Algorithm 1). Our training is customized for NLS. The heuristic learner is trained to minimize the expected TSP length of the NLS-refined sampled solutions.

## C Extended results

### C.1 DeepACO for more COP scales

We evaluate DeepACO on more COP scales and present the results in Fig. 9. The evolution curves demonstrate that DeepACO can provide consistent neural enhancement across COP scales.

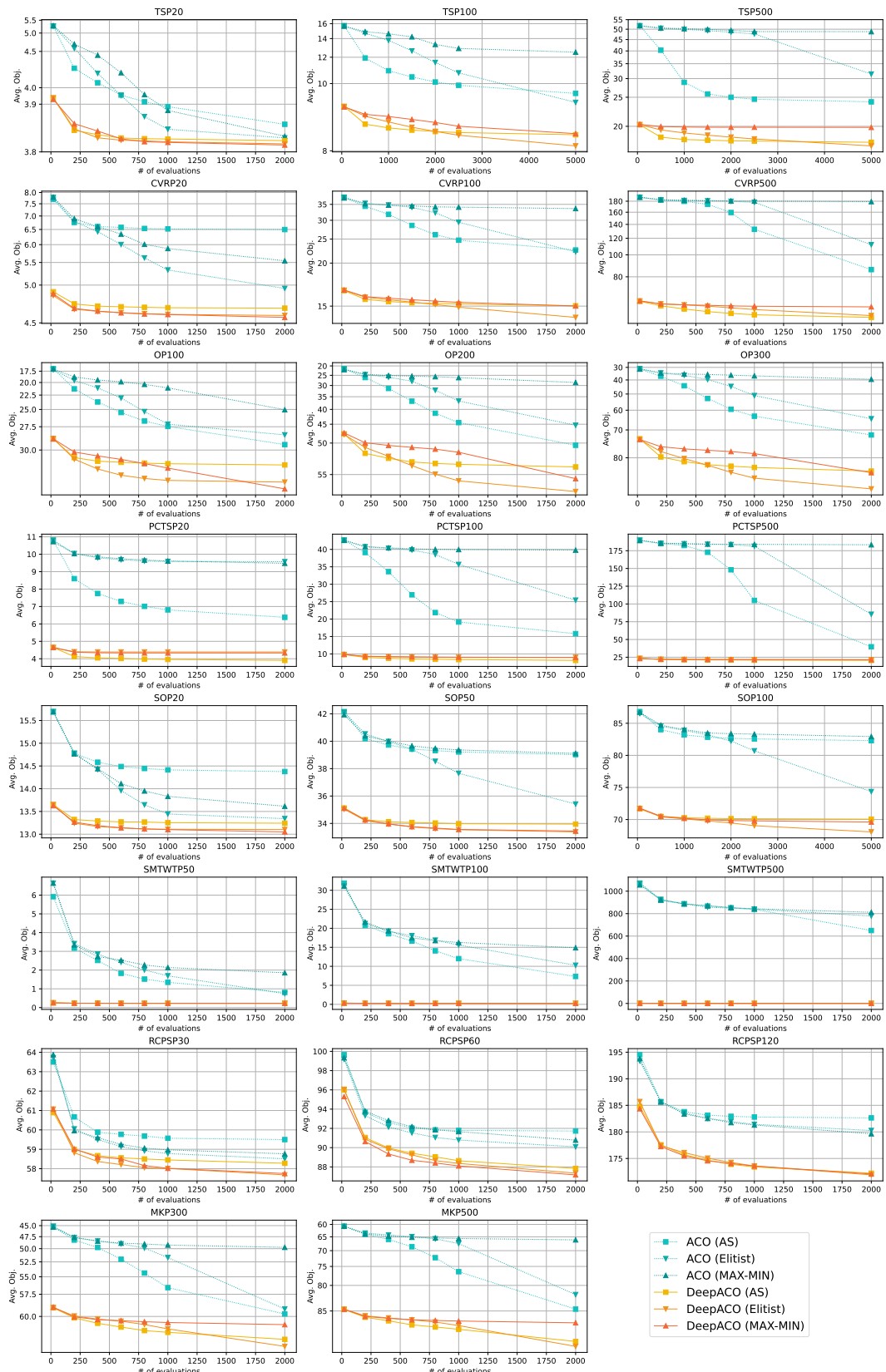

Figure 9: Evolution curves of fundamental DeepACO and ACO algorithms on eight different COPs with more scales. For each COP, we plot the best-so-far objective value (averaged over 100 held-out test instances) w.r.t. the number of used evaluations along the ACO iterations.

Table 3: Comparison results on TSP100.

| Method | TSP100 Obj. | TSP100 Time (s) |
|---|---|---|
| AM [54] | 7.945 | 0.36 |
| GCN [47] | 7.907 | 6.13 |
| da Costa et al. [22] | 7.821 | 30.66 |
| Hudson et al. [42] | 7.815 | 10.11 |
| Att-GCRN+MCTS [32] | 7.764 | 0.53 |
| DeepACO (NLS, T=4) | 7.767 | 0.50 |
| DeepACO (NLS, T=10) | 7.763 | 1.23 |

Table 4: The comparison results on CVRP.

| Method | CVRP100 Obj. | CVRP100 Time (s) | CVRP400 Obj. | CVRP400 Time (s) | CVRP1000 Obj. | CVRP1000 Time (s) | CVRP2000 Obj. | CVRP2000 Time (s) |
|---|---|---|---|---|---|---|---|---|
| AM [54] | 16.42 | 0.06 | 29.33 | 0.20 | 61.42 | 0.59 | 114.36 | 1.87 |
| TAM-LKH3 [41] | 16.08 | 0.86 | 25.93 | 1.35 | 46.34 | 1.82 | 64.78 | 5.63 |
| DeepACO (NLS,T=4) | 16.07 | 2.97 | 25.31 | 3.65 | 45.00 | 10.21 | 61.89 | 14.53 |
| DeepACO (NLS,T=10) | 15.77 | 3.87 | 25.27 | 5.89 | 44.82 | 15.87 | 61.66 | 35.94 |

## C.2 TSP and CVRP

Table 3 and 4 gather the comparative experimental results on more diverse routing tasks, i.e., TSP100, CVRP100, 400, 1000, and 2000. They demonstrate DeepACO's consistent and competitive performance. "Time" in the tables refers to the duration needed to solve a single instance, and the results of the baselines are drawn from their respective papers or [41]. NLS for CVRP is based on the local search strategy in the HGS-CVRP algorithm [80, 81].

**NLS for CVRP** The local search strategy proposed by Vidal [80] (denoted as $LS_{HGS}$) consists of searching among 5 collections of neighborhoods, including SWAP, RELOCATE, 2-opt, 2-opt*, and SWAP* neighborhood. NLS for CVRP follows Algorithm 1 and apply $LS_{HGS}$ as the local search strategy. In practice, we implement the code provided by Vidal [80] and set $T_{NLS}$ to 1. To be noted that $LS_{HGS}$ occasionally produces infeasible solutions, and such solutions are discarded in NLS. We refer the reader to [80] for more details.

## C.3 Generalizability of DeepACO

**Across COPs** The generalizability of DeepACO across COPs is rooted in the generalizability of the ACO meta-heuristic and the feasibility to represent many COPs' solutions using binary variables. In our paper, we evaluated DeepACO across 8 diverse COPs, including routing, assignment, scheduling, and subset COP types. Furthermore, we extend DeepACO to the Bin Packing Problem (BPP), a grouping problem, aiming at optimally splitting items into groups. We follow the experimental setup in [58] and demonstrate the consistent neural enhancement provided by DeepACO in Table 5.

**Across scales** Table 6 compares the performance of ACO, DeepACO trained on TSP100, and DeepACO trained on the respective test scale, all implementing vanilla LS (instead of NLS to ensure the same execution time for DeepACO and ACO). The results show that DeepACO still outperforms its ACO counterpart even with a significant distributional shift (i.e., from TSP100 to TSP1000).

Table 5: The evolution trend of ACO and DeepACO on the Bin Packing Problem.

| T | 1 | 5 | 10 | 20 | 30 | 40 |
|---|---|---|---|---|---|---|
| ACO ↑ | 0.877 | 0.896 | 0.902 | 0.907 | 0.909 | 0.910 |
| DeepACO ↑ | 0.947 | 0.952 | 0.954 | 0.956 | 0.957 | 0.958 |

Table 6: The generalization performance of DeepACO across TSP scales.

| Method | TSP500 | TSP1000 |
|---|---|---|
| ACO (LS) | 17.55 | 24.93 |
| DeepACO (LS, trained on TSP100) | 17.18 | 24.69 |
| DeepACO (LS) | 16.98 | 23.96 |

## C.4 Training duration

As presented in Table 7, DeepACO entails only minutes of training to provide substantial neural enhancement in Fig. 2. It is mainly because our training hybridizes 1) heatmap-based on-policy sampling without costly step-by-step neural decoding [66], and 2) parallel multi-start sampling leveraging the solution symmetries [56, 52].

Table 7: Training settings for DeepACO. "m" denotes minutes.

| COP | Problem scale | Total traninig instances | Total training time |
|---|---|---|---|
| TSP | 20 / 100 / 500 | 640 / 640 / 640 | 1m / 2m / 8m |
| CVRP | 20 / 100 / 500 | 640 / 640 / 640 | 2m / 4m / 18m |
| OP | 100 / 200 / 300 | 320 / 320 / 320 | 4m / 6m / 7m |
| PCTSP | 20 / 100 / 500 | 320 / 320 / 640 | 1m / 1m / 10m |
| SOP | 20 / 50 / 100 | 640 / 640 / 640 | 1m / 2m / 4m |
| SMTWTP | 50 / 100 / 500 | 640 / 640 / 640 | 1m / 2m / 11m |
| RCPSP | 30 / 60 / 120 | 3200 / 3200 / 3200 | 2m / 4m / 8m |
| MKP | 300 / 500 | 640 / 640 | 26m / 30m |

## C.5 Flexibility for a different pheromone model

The pheromone model used in ACO can vary, even for the same problem [63]. The pheromone model used in this work models the successive relationships of the nodes in a COP graph, which applies to most COPs but is by no means exhaustive. This section aims to show the flexibility of DeepACO for different pheromone models.

Here, we use the Multiple Knapsack Problem (MKP) (formally defined in Appendix D.8) as an example. In Section 5.2, we mostly use the pheromone model $PH_{suc}$ that models successively chosen items [63], where the pheromone trails are deposited on the edges connecting two items. One alternative, i.e., $PH_{items}$, is to directly model how good each item is, where the pheromone trails are deposited on the items, i.e., the graph nodes. On the one hand, $PH_{suc}$ is reasonable because the quality of an item must be considered in conjunction with how it is combined with other items. On the other hand, $PH_{items}$ makes sense because a solution to MKP is order-invariant. Both pheromone models have their own advantages [63] and are used in the recent literature [30, 8].

Isomorphic GNN [48] is applied in this work to $PH_{suc}$ due to its ability to extract edge features. To apply DeepACO to variations of pheromone models, it is necessary to use a neural model properly aligned with the pheromone model. As a showcase, we use a lightweight Transformer [79] encoder (without positional encoding) with an MLP decoder on top of it to model $PH_{items}$ for MKP. Under the same experimental setup as in Section 5.1, we compare DeepACO with its ACO counterparts using $PH_{items}$ for MKP. The results presented in Fig. 7 demonstrate that DeepACO using $PH_{items}$ can also outperform its ACO counterparts. It validates that DeepACO can be readily extended to variations of pheromone models using properly aligned neural models.

## C.6 Visualization of the learned heuristic

We visualize the learned heuristic in Fig. 10. After the graph sparsification (see Appendix D), we scatter the remaining solution components (i.e., the TSP edges in this case) in a unit square by normalizing (in a row-wise fashion) their manually designed and learned heuristic measures. We can observe that the solution components with a large learned heuristic measure do not necessarily

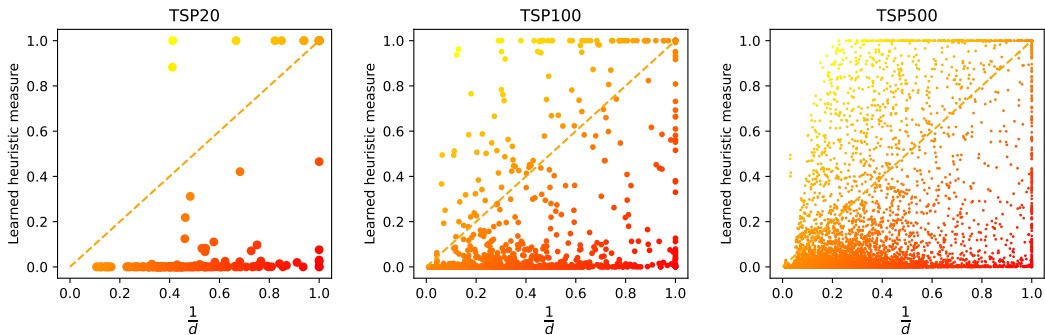

Figure 10: Visualization of the learned heuristic on TSP. We randomly sample a TSP instance $\rho$ and scatter its learned heuristic measure $h_{\boldsymbol{\theta}}(\boldsymbol{\rho})$ w.r.t. the manually designed ones, both after normalization. The dashed line represents an identical mapping.

obtain an equivalently large $\frac{1}{d}$, and vice versa. In particular, DeepACO learns a set of more aggressive heuristic measures, where most solution components are associated with normalized measures close to zero.

## D  Benchmark problems

### D.1  Traveling Salesman Problem

**Definition**   Traveling Salesman Problem (TSP) seeks the shortest route that visits a given set of nodes and returns to the starting node, visiting each node exactly once.

**Instance generation**   Nodes are sampled uniformly from $[0, 1]^2$ unit.

**Model inputs**   A TSP instance corresponds to a complete graph. Node features are their coordinates, and edge attributes are relative distances between nodes.

**Graph sparsification**   All nodes are only adjacent to their $k$ nearest neighbors to exclude highly unpromising solution components and speed up computation. The same graph sparsification is conducted for ACO baselines to ensure a fair comparison. $k$ is set to 10 for TSP20, 20 for TSP50, 20 for TSP100, and 50 for TSP500.

**Baseline heuristic**   The baseline heuristic measure is the inverse of edge length [72].

### D.2  Capacitated Vehicle Routing Problem

**Definition**   Capacitated Vehicle Routing Problem (CVRP) is a natural extension of TSP that involves multiple vehicles. In CVRP, each vehicle has a limited capacity, and each customer has its own demand that must be satisfied. The objective is to find the optimal route that satisfies all customer demands and minimizes the total distance traveled by the fleet.

**Instance generation**   We follow the same instance generation settings as in previous work [94]. Customer locations are sampled uniformly in the unit square; customer demands are sampled from the discrete set $\{1, 2, \ldots, 9\}$; the capacity of each vehicle is set to 50; the depot is located at the center of the unit square.

**Model inputs**   We transform a CVRP instance into a complete graph, where the node features are the demand values, and the edge attributes are relative distances between nodes.

**Baseline heuristic**   The baseline heuristic measure is the inverse of edge length [15].

### D.3 Orienteering Problem

**Definition** Following the problem setup in [54], each node $i$ has a prize $p_i$, and the objective of the Orienteering Problem (OP) is to form a tour that maximizes the total collected prize. The tour starts from a special depot node 0 and should be subject to a maximum length constraint.

**Instance generation** We uniformly sample the nodes, including the depot node, from the unit $[0,1]^2$. We use a challenging prize distribution [31]: $p_i = (1 + \left\lfloor 99 \cdot \frac{d_{0i}}{\max_{j=1}^{n} d_{0j}} \right\rfloor)/100$, where $d_{0i}$ is the distance between the depot and node $i$. The maximum length constraint is also designed to be challenging. As suggested by Kool et al. [54], we set it to 4, 5, and 6 for OP100, OP200, and OP300, respectively.

**Model inputs** We transform an OP instance into a complete graph. The node features are two-dimensional, consisting of nodes' prizes and distances to the depot. The edge attributes are relative distances between nodes. We do not include the maximum length constraint in input features because, in our case, it is a constant for each problem scale. It is easy to model the constraint explicitly as an additional node feature or context feature, or implicitly by using it to normalize nodes' distances to the depot.

**Graph sparsification** $k$ is set to 20 for OP100, 50 for OP200, and 50 for OP300.

**Baseline heuristic** The baseline heuristic measure $\eta_{ij} = \frac{p_j}{d_{ij}}$ [74].

### D.4 Prize Collecting Traveling Salesman Problem

**Definition** Prize Collecting Traveling Salesman Problem (PCTSP) is a variant of TSP. In PCTSP, each node has an associated prize and penalty. The objective is to minimize the total length of the tour (which visits a subset of all nodes) plus the sum of penalties for unvisited nodes while also ensuring a specified minimum total prize collected.

**Instance generation** We follow the design of Kool et al. [54] for instance generation. All nodes are sampled uniformly in the unit square. The prize for each node is sampled uniformly from $(0, 1)$. As a result, the minimum total prize constraint is set to $\frac{n}{4}$ for PCTSP$n$, such that $\frac{n}{2}$ nodes are expected to be visited. The penalty of each node is sampled uniformly from $(0, \frac{3L^n}{2n})$, where $L^n$ is the expected tour length for TSP$n$ and is set to 4, 8, 18 for $n = 20, 100, 500$, respectively. For an additional depot node, its prize and penalty are set to 0.

**Model inputs** We transform a PCTSP instance into a complete graph. The node features are two-dimensional, consisting of nodes' prizes and penalties. The edge attributes are relative distances between nodes.

**Baseline heuristic** PCTSP is related to OP but inverts the goal [54]. The baseline heuristic measure of edge $< i, j >$ is $\frac{p_j}{d_{ij}}$ [74].

### D.5 Sequential Ordering Problem

**Definition** The Sequential Ordering Problem (SOP) involves finding a consistent and linear ordering of a set of elements. Consider that we have $n$ jobs to handle sequentially for SOP$n$. Let $x_i$ be the processing time of job $i$ and $t_{ij}$ the waiting time between job $i$ and $j$. Specially, job 0 is a dummy node, and $x_0 = 0$. We also have a set of constraints that indicate the precedence between jobs. The objective is to minimize the processing time for all jobs under precedence constraints.

**Instance generation** We sample both $x_i$ and $t_{ij}$ uniformly from $(0, 1)$. We stochastically sample the precedence constraints for all job pairs while ensuring at least one feasible solution and maintaining the constraints' transitive property; if job $i$ precedes job $j$ and job $j$ precedes job $k$, then job $i$ precedes job $k$. The probability of imposing the constraint on a pair of jobs is set to 0.2.

**Model inputs** We construct a directed graph where each job is a node and each job pair $< i, j >$, as long as job $j$ does not precedes job $i$, is connected with a directed edge. The node feature is $x_i$ for job $i$ and the edge attribute for job pair $< i, j >$ is $t_{ij}$.

**Baseline heuristic** The baseline heuristic measure $\eta_{ij} = \frac{1}{x_j + d_{ij}}$ [70, 71].

### D.6 Single Machine Total Weighted Tardiness Problem

**Definition** The Single Machine Total Weighted Tardiness Problem (SMTWTP) is a scheduling problem in which a set of $n$ jobs must be processed on a single machine. Each job $i$ has a processing time $p_i$, a weight $w_i$, and a due date $t_i$. The objective is to minimize the sum of the weighted tardiness of all jobs, where the weighted tardiness is defined as the product of a job's weight and the duration by which its completion time exceeds the due date.

**Instance generation** Due dates, weights, and processing time are sampled as follows: $t_i \sim n \times Uni(0, 1), w_i \sim Uni(0, 1), p_i \sim Uni(0, 1)$.

**Model inputs** To ensure that all jobs can be the first to get processed, we include a dummy job with all its features set to zero. Then, we transform an SMTWTP instance into a directed complete graph with $n + 1$ nodes. The features of a node are its normalized due date and weight. The attribute of edge $< i, j >$ is $p_j$.

**Baseline heuristic** The baseline heuristic prefers jobs with shorter due time: $\eta_{ij} = \frac{1}{t_j}$ [1].

### D.7 Resource-Constrained Project Scheduling Problem

**Definition** Resource-Constrained Project Scheduling Problem (RCPSP) is to schedule a set of activities and minimize the schedule's makespan while satisfying precedence and resource constraints.

**Instances** Due to the complexity of generating random RCPCP instances, we draw the instances from PSPLIB [53] for training and held-out test.

**Model inputs** As stated by Gonzalez-Pardo et al. [36], parallelism is not allowed if we sample directly from the constraint graph. So the input graph is an extension of it by adding edges between unrelated nodes, i.e., the edge between node $i$ and $j$ ($i \neq j$) exists only if $j \notin \mathcal{S}_i^* \wedge i \notin \mathcal{S}_j^*$, where $\mathcal{S}_i^*$ denotes all the successors (direct and indirect) of node $i$ on the constraint graph.

**ACO algorithm** The ACO algorithm follows the architecture proposed by Merkle et al. [62] but without the additional features. We sample the ant solutions in a topological order to accelerate the convergence.

**Baseline heuristic** The baseline heuristic is the combination of the greatest rank positional weight all (GRPWA) heuristic, and the weighted resource utilization and precedence (WRUP) heuristic [62].

### D.8 Multiple Knapsack Problem

**Definition** Knapsack Problem (KP) is to select a subset of items from a given set such that the total weight of the chosen items does not exceed a specified limit and the sum of their values is maximized. Multiple Knapsack Problem (MKP), also known as $m$-dimensional Knapsack Problem, impose $m$ different constraints on $m$ types of weight. Formally, MKP can be formulated as follows:

$$
\begin{aligned}
& \text{maximize} \sum_{j=1}^{n} v_j x_j \\
& s.t. \sum_{j=1}^{n} w_{ij} x_j \leq c_i, \quad i = 1, \ldots, m, \\
& \quad x_j \in \{0, 1\}, \quad j = 1, \ldots, n.
\end{aligned}
\tag{11}
$$

Here, $v_j$ is the value of the item $j$, $w_{ij}$ is the $i$-th type of weight of the item $j$, and $x_j$ is the decision variable indicating whether the item $j$ is included in the knapsack.

**Instance generation**  The values and weights are uniformly sampled from $[0, 1]$. To make all instances well-stated [57], we uniformly sample $c_i$ from $(\max_j w_{ij}, \sum_j w_{ij})$.

**Model inputs**  For $PH_{suc}$, we transform an MKP instance into a complete directed graph. The node features are their weights. The attribute of edge $< i, j >$ is $v_i$. For $PH_{items}$, the model input is $m + 1$-dimensional feature vectors of $n$ items. The feature vector of item $j$ is a concatenation of $v_j$, and $w_{ij}, i = 1, \ldots, m$ normalized by dividing $c_i, i = 1, \ldots, m$ respectively.

**Baseline heuristic**  For $PH_{suc}$, the baseline heuristic $\eta_{ij} = \frac{v_j}{\sum_k w_{kj}}$ [30]. For $PH_{items}$, the baseline heuristic $\eta_j = \frac{v_j}{\sum_k w_{kj}}$ [30].

### D.9  Bin Packing Problem

**Definition**  The objective of the Bin Packing Problem (BPP) is to optimally arrange items within containers of a specific capacity, in order to reduce the overall number of containers used. We follow the specific setup in [58].

**Instance generation**  Following [58], we set the number of items to 120 and the bin capacity to 150. Item sizes are uniformly sampled between 20 and 100.

**Model inputs**  We transform a BPP instance into a complete graph where each node corresponds to an item. Node features are their respective item sizes, while edge features are set to 1. We additionally include a dummy node with 0 item size as an unbiased starting node for solution construction.

**Baseline heuristic**  Heuristic measure of an item equals its size [58]; that is, the heuristic measure of a solution component in $PH_{suc}$ is the item size of its ending node.

