# OpenReview forum: "DeepACO: Neural-enhanced Ant Systems for Combinatorial Optimization"
_NeurIPS.cc/2023/Conference — NeurIPS 2023 poster_

### Official Review · Reviewer_d7BW · 2023-06-30

**Soundness:** 4 excellent
**Presentation:** 4 excellent
**Contribution:** 3 good
**Rating:** 6
**Confidence:** 4

**Summary:**

This paper proposes a method to improve the performance of existing meta-heuristic algorithms using deep neural networks. At the same time, the method described in this paper makes it possible to design a high-performing meta-heuristic without requiring expert domain knowledge. In particular, the study in this paper is generally applicable to many types of COPs. This paper presents a deep neural network model to output the heuristic measure of ACO, as well as a training method for it. In addition, a neural-guided pertubating method is presented in the local search process. Experiments were conducted on 8 different types of CO problems, which showed that DeepACO significantly improves the performance of conventional ACO. It also showed excellent results when compared with the recent NCO studies. Finally this paper is excellently written and well organized.

**Strengths:**

**S1.** The method of combining meta-heuristic and deep neural networks is novel and applicable to various types of CO problems.

**S2.** The performance of the ACO meta-heuristic has been greatly improved by deepACO. In addition, the experiments were appropriately designed and performed.

**S3.** Compared to recent studies on NCO, DeepACO's performance was shown to be superior.


**Weaknesses:**

**W1.** In section 5.3, it is difficult to conclusively state that DeepACO demonstrates state-of-the-art performance when compared to the latest NCOs, because only the TSP500 and TSP1000 tasks are being used for comparison with existing NCOs. It would be nice to have comparative experiment results with existing NCOs for more diverse tasks. For example, adding CVRP 500/1000 experiments or adding TSP 100/CVRP 100, etc.

**W2.** In section 4.2, details about neural-guided perturbation are lacking. Specifically, there is a lack of explanation regarding the structure of the neural network used for perturbation, as well as details on the input and output of the network, and the processes of learning and inference.

​

**Questions:**

**Q1.** Please explain in more detail about neural-guided perturbation in 4.2. (refer to W2)

**Q2.** Since $\tau_{i,j}$ is fixed to $1$ when training a heuristic learner(line 183), $\tau$ is not considered in the training process. However, in Figure 4, DeepACO is robust against changes in alpha. What makes DeepACO robust to changes in alpha?

**Q3.** In line 252, you wrote "We extend DeepACO from $PH_{suc}$ to $PH_{items}$ using a Transformer encoder equipped with an MLP decoder". This is a new neural network different from the heuristic learner described in 4.3. Please explain in detail about this neural network.

**Q4.** How many TSP test instances were used in the experiment in Table 1 and Table 2?

---

> ### Author Rebuttal · Authors · 2023-08-08
>
> # Response to Reviewer d7BW
> We appreciate your insightful comments and constructive suggestions. Below we present our point-to-point response.
>
> ---
>
> ### Response to Weaknesses
>
> > W1. It would be nice to have comparative experiment results with existing NCOs for more diverse tasks, e.g., CVRP 500/1000 or TSP 100/CVRP 100.
>
> Thank you for raising this concern. We have added the suggested experiments and presented the results in our **Global Response**. They demonstrate DeepACO's competitive performance across diverse combinatorial optimization tasks (TSP100, CVRP100, 400, 1000, and 2000).
>
> > W2. In section 4.2, details about neural-guided perturbation are lacking, i.e., the structure of the neural network, its input and output, and its processes of learning and inference.
>
> Neural-guided perturbation does not involve a new neural network other than that introduced in Section 4.1. It directly leverages the learned heuristic measures $\eta_{\theta}$ to guide the perturbation process. Specifically, each time a solution reaches local optima, it gets perturbed with local search (LS) which iteratively maximizes the heuristic measures of its solution components. Note that NLS can be implemented using an arbitrary LS operator.
>
> Take TSP as an example. With 2-opt as the LS operator, NLS alternates between (a) optimizing a solution to minimize its tour length and (b) perturbing it to maximize the total heuristic measures of its edges. The two processes are similar, except that we use the inverse of heuristic measures in (b) while the distance matrix in (a) as indicators of good edges (line 6 in Algorithm NLS). Our training is customized for NLS. The heuristic learner is trained to minimize the expected TSP length of the NLS-refined sampled solutions.
>
> Regarding the heuristic learner introduced in Section 4.1, it utilizes the neural architecture described in lines 155-158 and Appendix A. The inputs and outputs of this learner are COP graphs as detailed in Appendix C, and learned heuristic measures, respectively. Its training process is described in Section 4.3, and it follows the ACO algorithm introduced in Section 3 during inference.
>
> ---
>
> ### Response to Questions
>
> > Q1. Please explain in more detail about neural-guided perturbation in 4.2.
>
> Please refer to our response to W2.
>
> > Q2. What makes DeepACO robust to changes in alpha?
>
> Thank you for this insightful question. Traditionally, $\alpha$ is tuned to balance pheromones and heuristics for controlling convergence speed. Both premature convergence and slow convergence could lead to poor solutions. In DeepACO, the learned heuristic measures already provide a close-to-optimal initial search point. As a result, controlling convergence speed becomes less critical.
>
> > Q3. Please explain in detail about the neural network used for $PH_{items}$.
>
> The neural architecture used for $PH_{items}$ is detailed below and will be added to our paper.
>
> The architecture mostly follows the transformer encoder (num_hidden_layers=3, hidden_size=32, num_attention_heads=2) but deprecates its positional encoding. On top of it, we add position-wise feedforward layers (num_hidden_layers=3, hidden_size=32) that map the hidden representations of each solution component into its real-valued heuristic measure. Overall, it is similar to the neural networks used in [1,2].
>
> > Q4. How many TSP test instances were used in the experiment in Table 1 and Table 2?
>
> As in previous works, we utilized the datasets released in [3], each comprising 128 test instances. This information will be added to the paper.
>
> ---
>
> **References**
>
> [1] Devlin, J., Chang, M. W., Lee, K., & Toutanova, K. (2018). Bert: Pre-training of deep bidirectional transformers for language understanding. arXiv preprint arXiv:1810.04805.
>
> [2] Goh, Y. L., Lee, W. S., Bresson, X., Laurent, T., & Lim, N. (2022). Combining reinforcement learning and optimal transport for the traveling salesman problem. arXiv preprint arXiv:2203.00903.
>
> [3] Fu, Z. H., Qiu, K. B., & Zha, H. (2021). Generalize a small pre-trained model to arbitrarily large TSP instances. In Proceedings of the AAAI conference on artificial intelligence (Vol. 35, No. 8, pp. 7474-7482).

---

> > ### Comment · Reviewer_d7BW · 2023-08-14
> >
> > Thank you for the detailed responses and additional experiments. I have no further questions. I will keep the rating unchanged.

---

> > > ### Author Response · Authors · 2023-08-14
> > > **Thanks for reviewing**
> > >
> > > We sincerely appreciate your thoughtful review and valuable feedback.

---

### Official Review · Reviewer_BitH · 2023-07-06

**Soundness:** 3 good
**Presentation:** 3 good
**Contribution:** 3 good
**Rating:** 5
**Confidence:** 4

**Summary:**

This paper proposes DeepACO, a generic framework leveraging deep reinforcement learning to automate heuristic designs. Specifically, DeepACO serves to strengthen the heuristic measures of existing ACO algorithms and dispense with laborious manual design in future ACO applications.  Experiments demonstrate that DeepACO consistently outperforms its ACO counterparts on eight combinatorial optimization problems (COPs).

**Strengths:**

1. To the best of my knowledge, this work is the first to exploit deep reinforcement learning to guide the evolution of ACO meta-heuristics.
2. Experiments demonstrate that DeepACO consistently outperforms its ACO counterparts on eight combinatorial optimization problems (COPs).


**Weaknesses:**

1. Previous work [1] has proposed a data-driven Heuristics Schedule framework in modern mixed-integer linear programming (MILP) solvers. The authors may want to discuss the major novelty of this work over previous work.
2. It would be more convincing if the authors could show some generalization results, as the ability of generalizing to larger instances is also important in solving MILPs.

[1] Chmiela, Antonia, et al. "Learning to schedule heuristics in branch and bound." Advances in Neural Information Processing Systems 34 (2021): 24235-24246.


**Questions:**

Please refer to Weaknesses for my questions.

**Limitations:**

Yes.

---

> ### Author Rebuttal · Authors · 2023-08-08
>
> # Response to Reviewer BitH
>
> Thank you for your time and effort in reviewing our submission, and we appreciate the fresh perspective you've offered. Below, we provide a thorough point-to-point response, trying to address each of your remarks.
>
> > W1: The authors may want to discuss the major novelty of this work over previous work [1].
>
> The two works tackle different problem domains with different goals, learned components, and training methodologies. DeepACO provides a flexible neural enhancement framework for ACO, while [1] optimizes heuristic scheduling within a branch and bound solver. The techniques are largely complementary.
>
> The key differences between the two works are as follows.
>
> - **Problem domains.** DeepACO focuses on combinatorial optimization and is based on ACO metaheuristics. In contrast, [1] targets mixed integer programming problems and operates within a branch and bound solver framework.
>
> - **Complementary goals.** Many combinatorial optimization problems can be formulated as Mixed Integer Linear Programs. MILP is a powerful tool for solving them exactly, especially for moderate-sized instances. However, for large, complex, and even black-box problems, using MILP may become infeasible. In such cases, (meta-)heuristics can be used to find good-quality solutions efficiently.
>
> - **Learned components.** DeepACO learns heuristic measures to guide the ACO construction process. [1] learns ordering and iteration limits for primal heuristics during branch and bound.
>
> - **Training methodology.** DeepACO leverages deep reinforcement learning to train neural networks across problem instances. [1] formulates training as an optimization problem using collected heuristic performance data.
>
> [1] Chmiela, Antonia, et al. "Learning to schedule heuristics in branch and bound." Advances in Neural Information Processing Systems 34 (2021): 24235-24246.
>
> > W2: It would be more convincing if the authors could show some generalization results.
>
> Thank you for bringing this to our attention. In Fig. 5, we demonstrate that DeepACO can generalize across scales and distributions while preserving its neural enhancement. Moreover, additional generalization results are presented in our **Global Response**, where we effectively generalize DeepACO trained on TSP100 to TSP1000, and extend DeepACO to even larger-scale CVRP.

---

> > ### Comment · Reviewer_BitH · 2023-08-16
> > **Thanks for the response**
> >
> > Thanks for the response. I have read the rebuttal and the other reviews. I have no further questions.

---

> > > ### Author Response · Authors · 2023-08-16
> > > **Thanks for reviewing**
> > >
> > > We sincerely appreciate your thoughtful review and valuable feedback.

---

### Official Review · Reviewer_YjbH · 2023-07-06

**Soundness:** 3 good
**Presentation:** 3 good
**Contribution:** 2 fair
**Rating:** 6
**Confidence:** 3

**Summary:**

This paper presents DeepACO, a neural-enhanced solution to the limitations of Ant Colony Optimization (ACO) meta-heuristics, namely, the laborious manual design of heuristic measures and their heavy reliance on expert knowledge. ACO, which is a foraging system inspired by ant colonies, deploys artificial ants to explore solution spaces, and these solutions are biased toward promising areas through pheromone trails and heuristic measures. DeepACO innovates by learning a problem-specific mapping from an instance to its heuristic measures, leveraging neural models across instances, and incorporating these measures into ACO to influence solution constructions and help escape local optima. The paper also proposes three extended implementations of DeepACO to balance exploration and exploitation. DeepACO outperforms traditional ACO methods and is competitive against state-of-the-art and problem-specific Neural Combinatorial Optimization (NCO) methods across eight combinatorial optimization problems, even with only minutes of training.

**Strengths:**

Novelty: The paper proposes DeepACO, a new approach to enhancing Ant Colony Optimization (ACO) algorithms through a neural-enhanced meta-heuristic. This application of deep reinforcement learning to ACO is stated as being the first of its kind, thereby representing an innovative contribution to the field.

Broad Applicability: DeepACO has been tested across eight different Combinatorial Optimization Problems (COPs), indicating its versatility and adaptability. It also shows competitive performance with specialized Neural Combinatorial Optimization (NCO) methods on the Travelling Salesman Problem (TSP), suggesting it is a robust method.

Automation: The paper presents a method to automate the design of heuristics for future ACO applications, potentially saving significant manual effort and expert knowledge.

**Weaknesses:**

Limitations in Current Implementation: The authors themselves note that DeepACO may underperform when not incorporating local search (LS) components due to the current restriction of compressing all learned heuristic information into an n × n matrix. This could limit the solution's effectiveness in complex COPs.

Absence of Comparison to Other Methods: While the paper mentions that DeepACO outperforms other ACO methods and is competitive with specialized NCO methods, there's no mention of a comparison with other non-ACO or non-NCO optimization methods. This could limit the ability to gauge how novel or effective DeepACO truly is in the broader context of optimization techniques.

Dependence on Machine Learning: The performance of DeepACO appears to be heavily dependent on machine learning. While this isn't necessarily a weakness, it could limit its application in situations where computational resources are restricted, or where machine learning models are difficult to train due to lacking training data.

Unclear Generalizability: Although DeepACO is tested on eight different COPs, it's unclear how it would perform on a broader range of problems, particularly those that don't closely resemble the tested ones. More comprehensive testing across a diverse set of problems and datasets would provide stronger evidence of its generalizability.

**Questions:**

Could you provide further insight into the "3D heuristic measures" or "dynamic heuristics" you mention in the conclusion? How might they help to overcome the limitations of the current implementation?

How do you see DeepACO being applied in real-world situations? What kind of problems do you envision it solving most effectively?

Could you discuss more about the process and challenges of training the deep reinforcement learning (DRL) model for guiding the evolution of ACO meta-heuristics?

---

> ### Author Rebuttal · Authors · 2023-08-08
>
> # Response to Reviewer YjbH
>
> We appreciate the time and effort you have dedicated to reviewing our submission. Your comments have provided us with a fresh perspective and have undoubtedly enhanced the quality of our work. Below we present our point-to-point response.
>
> ---
>
> ## Response to Weaknesses
>
> > W1: Limitations in Current Implementation: The authors themselves note that DeepACO may underperform when not incorporating local search (LS) components.
>
> We acknowledge the limitation of compressing learned heuristics into an n $\times$ n matrix. It is a limitation inherited from ACO as well as other constructive (meta-)heuristics for COPs. Since we directly sample solutions on this matrix (probabilistic graph), addressing this limitation leads to constructing higher-quality COP solutions in one shot with O(n) complexity. It is an exciting direction for future work, and we present several possible avenues toward it in our response to Q1.
>
> > W2: Absence of Comparison to non-ACO/NCO Methods
>
> We appreciate the reviewer highlighting the opportunity for comparison with non-ACO/NCO methods. As this work focuses on enhancing ACO algorithms with NCO techniques, we believe comparisons within these two domains are most relevant and fair. We agree further comparisons would provide a more holistic evaluation, and have added comparisons with Guided Local Search in our Global Response, showing DeepACO's superior performance on TSP.
>
>
> > W3: Dependence on Machine Learning: demanding computational resources and training data.
>
> We believe the reasons below can help address your concern and make DeepACO widely accessible.
>
> - **Low computational demand.** DeepACO only requires **minutes** of training (Appendix B.2) and a lightweight model (~50k parameters) to provide substantial neural enhancement. Moreover, we observed that training DeepACO solely on a **CPU**, rather than utilizing a GPU, even accelerates the training process on our hardware. This expedited training is primarily because most of the computational time is spent on on-policy solution sampling, rather than on the forward and backward neural network propagation.
>
> - **Minimal reliance on (real-world) training data.** DeepACO requires only **a few hundred training instances** in most tasks to achieve significant neural enhancement (Appendix B.2). Furthermore, when obtaining a realistic data distribution is challenging, we can confidently train DeepACO on algorithmically generated (Appendix C) **inexhaustible synthetic data** and generalize it effectively. Fig. 5 shows that DeepACO trained on fixed-scale and uniform synthetic data can clearly outperform its ACO counterparts on variable-scale real-world benchmarks.
>
> > W4: Unclear Generalizability: More comprehensive testing across a diverse set of problems and datasets would provide stronger evidence of its generalizability.
>
> The generalizability of DeepACO is rooted in the generalizability of the ACO meta-heuristic and the feasibility to represent many COPs' solutions using binary variables. In our paper, we evaluated DeepACO on **26 datasets** (please also refer to Appendix B.1 for more details) spanning **8 diverse COPs**, including routing, assignment, scheduling, and subset COP types, which do not resemble one another. Furthermore, DeepACO can be extended to an even broader range of COPs (e.g., Karp's 21 NP-complete problems), as exemplified in our general response.
>
> ---
>
> ## Response to Questions
>
> > Q1: further insight into the "3D heuristic measures" or "dynamic heuristics"
>
> "3D heuristic measures" and "dynamic heuristics" are possible extensions of the current 2D heuristic matrix, allowing for compressing more learned heuristics of a COP. It is possible that we can realize such extensions in various ways, where two possibilities are described below.
>
> - One option is to assign each ant its unique heuristic matrix and collaboratively train the ant population using individual-specific "niching loss" (inspired by the niching methods in Evolutionary Computation). In this manner, we obtain 3D heuristic measures, allowing for cooperatively exploring multiple optima of complex COPs or the Pareto front of multi-objective COPs.
>
> - Another possible approach, i.e., dynamic heuristics, generates heuristic measures at various points either throughout the solution-constructing process or during ACO iterations. In the first scenario, heuristic measures can be generated based on partial solutions. In the second scenario, we can learn to adapt the heuristic measures according to the updated pheromone trails.
>
> > Q2: How do you see DeepACO being applied in real-world situations? What kind of problems do you envision it solving most effectively?
>
> The ACO metaheuristic already has broad applicability in real-world situations [1]. DeepACO takes it one step further by dispensing with the required expert knowledge and automatically enhancing its performance, thereby extending its applications to scenarios involving more complex and black-box COPs. We believe it is particularly competitive for problems with little or suboptimal expert knowledge, as well as when instances to solve follow similar distributions.
>
> [1] Dorigo, M., & Stützle, T. (2019). *Ant colony optimization: overview and recent advances* (pp. 311-351). Springer International Publishing.
>
>
> > Q3: the process and challenges of training the deep reinforcement learning (DRL) model
>
> To train a DRL model for ACO, we need to (1) build a simulation environment based on COP constraints and ACO algorithm, (2) determine the distribution for sampling synthetic instances or gather real-world data, (3) set the RL reward based on COP objective, and (4) code the RL algorithm. Overall, DeepACO is relatively easy to train. Most coding effort was spent on building the simulation environment for each problem to enable efficient parallel solution sampling. By comparison, less effort was spent on the DRL training part, which is generic for different problems.

---

> > ### Comment · Area_Chair_B8h2 · 2023-08-17
> > **Follow up**
> >
> > Dear Reviewer,
> >
> > We would appreciate if you would you be so kind as to acknowledge and respond to the authors' rebuttal. This is crucial to ensure the reviewing process is conducted adequately.
> >
> > AC

---

> > ### Comment · Reviewer_YjbH · 2023-08-18
> >
> > The authors have done a good job to address my concerns. I have adjusted my score accordingly.

---

> > > ### Author Response · Authors · 2023-08-18
> > > **Thanks for reviewing**
> > >
> > > We sincerely appreciate your thoughtful review and valuable feedback.

---

### Official Review · Reviewer_hKqX · 2023-07-06

**Soundness:** 4 excellent
**Presentation:** 4 excellent
**Contribution:** 2 fair
**Rating:** 6
**Confidence:** 4

**Summary:**

This article proposes DeepACO, which is a generic framework leveraging deep reinforcement learning to automate heuristic designs.
DeepACO serves to strengthen the heuristic measures of existing ACO algorithms.
According to the experiments, DeepACO consistently outperforms its ACO counterparts on eight COPs using a single neural model and a single set of hyperparameters. It also performs better than or competitively against the problem-specific methods on the canonical Travelling Salesman Problem.

The article reviews the related works, explains ACO and the proposed methodology, and presents the experiments: their settings results and analysis. It is followed by supplementary materials.


**Strengths:**

It is a well-written paper and I enjoyed reading it. All important concepts seem to be clearly explained.
The method was tested in 8 benchmark problems which is a huge strength. On all of them, DeepACO outperformed standard ACO methods.
In general, the results of the experiments are very good. The limitations of the method are discussed as well and the authors declared that the code used in experiments will be made publicly available and I found it in supplementary materials, indeed.


**Weaknesses:**

For some reason, hyperlinks to the bibliography do not work in the PDF that I received. I didn't find information on how the parameters alpha and beta (control parameters) were set in experiments. The writing can be slightly improved:
- p.7: "On the other hand" appears at the beginning of 2 consecutive sentences, so maybe in 1 of those sentences, it can be substituted.

**Questions:**

Question to Fig. 6: does it make sense to combine all the extensions and use them at once?

**Limitations:**

The authors adequately discussed the limitations.

---

> ### Author Rebuttal · Authors · 2023-08-08
>
> # Response to Reviewer hKqX
>
> To begin with, we are encouraged that you enjoyed reading our paper and we sincerely appreciate your insightful feedback. Below, we provide our point-to-point response.
>
> > W1: Hyperlinks to the bibliography do not work.
>
> We sincerely regret any inconvenience caused by the hyperlink issue. We'll ensure the hyperlinks to the bibliography work in the final version.
>
> > W2: Missing information on how the parameters alpha and beta (control parameters) were set.
>
> Thank you for pointing this out. We've consistently set alpha and beta as 1 during our experiments, except when testing for hyperparameter robustness in Fig. 4. This setting will be incorporated into the final edit.
>
> > W3: writing can be slightly improved on p.7.
>
> Thank you for this suggestion. We will rephrase the sentence to avoid repetition.
>
> > Q: In Fig. 6, does it make sense to combine all the extensions and use them at once?
>
> Thank you for this insightful question. Yes, we can combine them since they are not mutually exclusive. However, combining all of them may not be more effective than implementing just one. This is because they are designed for a similar purpose, and a combination entails tuning more hyperparameters for effective training.

---

> > ### Comment · Reviewer_hKqX · 2023-08-10
> >
> > Thanks for the information. I've read the rebuttal and don't have more questions. As for now, I keep my current rating.

---

> > > ### Author Response · Authors · 2023-08-11
> > > **Thanks for reviewing**
> > >
> > > We sincerely appreciate your thoughtful review and valuable feedback.

---

### Author Rebuttal · Authors · 2023-08-08

# Global Response

We are grateful to the reviewers for their insightful feedback and for recognizing the merit of our paper, e.g., novelty (Reviewer YjbH, BitH, d7BW), generalizability (Reviewer hKqX, YjbH, d7BW), effectiveness (Reviewer YjbH, BitH, d7BW), excellent presentation (Reviewer hKqX, d7BW). We have tried our best to address your primary concerns and will also rectify all minor issues raised.

In our paper, we evaluated DeepACO on 26 datasets (please also refer to Appendix B.1) spanning 8 diverse COPs, including routing, assignment, scheduling, and subset COP types. To further demonstrate the generalizability and superiority of DeepACO, we perform additional experiments with more diverse tasks/scales and introduce more baselines. Before responding to each reviewer's specific comments, we present additional experimental results here.

---

(*Review YjbH-W2*) The table below compares DeepACO with Guided Local Search, a non-ACO/NCO metaheuristic used in Google OR tools, and reported to be “generally the most efficient metaheuristic for vehicle routing [1]”. The results show DeepACO's superior performance on TSP.

||TSP100||TSP500||TSP1000||
|-|-|-|-|-|-|-|
||Obj.|Time|Obj.|Time|Obj.|Time|
|Guided Local Search|7.83|10s|17.32|20s|24.29|40s|
|DeepACO (ours)|7.76|1.2s|16.86|10s|23.85|32s|

---

(*Review YjbH-W4*) The table below showcases that DeepACO can be extended to an even broader range of COPs. Specifically, we additionally tackle Bin Packing Problem (BPP), a grouping problem, aiming to optimally split items into groups. We follow the experimental setup described in [2] and demonstrate DeepACO’s consistent neural enhancement.

T|1|5|10|20|30|40
-|-|-|-|-|-|-
ACO $\uparrow$|0.877|0.896|0.902|0.907|0.909|0.910
DeepACO $\uparrow$|0.947|0.952|0.954|0.956|0.957|0.958

---

(*Reviewer BitH-W2*) The table below compares the performance of ACO, DeepACO trained on TSP100, and DeepACO trained on the respective test scale, all implementing vanilla LS (instead of NLS to ensure the same execution time for DeepACO and ACO). The results show that DeepACO still outperforms its ACO counterpart even with a significant distributional shift (i.e., from TSP100 to TSP1000).

||TSP500|TSP1000|
|-|-|-|
ACO (LS)|17.55|24.93|
|DeepACO (LS, trained on **TSP100**)|17.18|24.69|
|DeepACO (LS)|16.98|23.96|

---

(*Reviewer d7BW-W1, BitH-W2*) The tables below gather the comparative experimental results with existing NCO methods for more diverse tasks, i.e., CVRP100, 400, 1000, 2000 and TSP100. They demonstrate DeepACO's consistent and competitive performance. The NLS strategy for CVRP is based on the SWAP* neighborhood [3].

|                   | CVRP100      | CVRP400      | CVRP1000      | CVRP2000      |
| -----             | -----        | -----        | -----         | -----         |
| AM [4]            | 16.42(0.06s) | 29.33(0.20s) | 61.42(0.59s)  | 114.36(1.87s) |
| TAM-LKH3 [5]      | 16.08(0.86s) | 25.93(1.35s) | 46.34(1.82s)  | 64.78(5.63s)  |
| DeepACO(NLS,T=4)  | **16.07**(2.97s) | **25.31**(3.65s) | **45.00**(10.21s) | **61.89**(14.53s) |
| DeepACO(NLS,T=10) | **15.77**(3.87s) | **25.27**(5.89s) | **44.82**(15.87s) | **61.66**(35.94s) |

|                     | TSP100        |
| -----               | -----         |
| AM [4]              | 7.945(0.36s)  |
| GCN [6]             | 7.907(6.13s)  |
| da Costa et al. [7] | 7.821(30.66s) |
| Hudson et al. [8]   | 7.815(10.11s) |
| Att-GCRN+MCTS [9]   | 7.764(0.53s)  |
| DeepACO(NLS,T=4)    | 7.767(0.50s)  |
| DeepACO(NLS,T=10)   | **7.763**(1.23s)  |

---

**References**

[1] Google. Google or-tools.

[2] Levine, J., & Ducatelle, F. (2004). Ant colony optimization and local search for bin packing and cutting stock problems. Journal of the Operational Research Society, 55(7), 705-716.

[3] Vidal, T. (2022). Hybrid genetic search for the CVRP: Open-source implementation and SWAP* neighborhood. *Computers & Operations Research*, *140*, 105643.

[4] Kool, W., van Hoof, H., & Welling, M. (2019). Attention, Learn to Solve Routing Problems! (arXiv:1803.08475). arXiv.

[5] Hou, Q., Yang, J., Su, Y., Wang, X., & Deng, Y. (2023). Generalize Learned Heuristics to Solve Large-scale Vehicle Routing Problems in Real-time. The Eleventh International Conference on Learning Representations.

[6] Joshi, C. K., Laurent, T., & Bresson, X. (2019). An Efficient Graph Convolutional Network Technique for the Travelling Salesman Problem (arXiv:1906.01227). arXiv.

[7] Costa, P. R. d O., Rhuggenaath, J., Zhang, Y., & Akcay, A. (2020). Learning 2-opt Heuristics for the Traveling Salesman Problem via Deep Reinforcement Learning. Proceedings of The 12th Asian Conference on Machine Learning, 465–480.

[8] Hudson, B., Li, Q., Malencia, M., & Prorok, A. (2022). Graph Neural Network Guided Local Search for the Traveling Salesperson Problem (arXiv:2110.05291). arXiv.

[9] Fu, Z. H., Qiu, K. B., & Zha, H. (2021). Generalize a small pre-trained model to arbitrarily large TSP instances. In Proceedings of the AAAI conference on artificial intelligence (Vol. 35, No. 8, pp. 7474-7482).

---

### Decision · Program_Chairs · 2023-09-21

**Decision:**

Accept (poster)

**Comment:**

This paper proposes ways to use deep RL algorithms for automating the heuristic designs of the ACO (Ant Colony Optimization) algorithms for Combinatorial Optimization Problems (COPs). The authors then show via experiments their algorithm DeepACO performs better than the heristic baselines in the Traveling Salesman Problem. Although tepidly the reviewers agreed on the value and novelty of this work. There was good amount of enthusiasm for the experimental results shown in the paper.